# Chemical Peeling Therapy Using Phenol for the Cervico-Vaginal Intraepithelial Neoplasia

**DOI:** 10.3390/v15112219

**Published:** 2023-11-07

**Authors:** Toshiyuki Maehama, Sumire Shimada, Jinichi Sakamoto, Takeo Shibata, Satoko Fujita, Masahiro Takakura, Hiroaki Takagi, Toshiyuki Sasagawa

**Affiliations:** 1Department of Obstetrics and Gynecology, Yuai-Medical Center, Tomishiro 901-0224, Okinawa, Japan; tmaehama@yuuai.or.jp; 2Department of Obstetrics and Gynecology, Kanazawa Medical University, 1-1 Daigaku, Kahoku-gun 920-0293, Ishikawa, Japan

**Keywords:** human papillomavirus, CIN, chemical-peeling therapy

## Abstract

**Objective:** This study aimed to validate the use of liquid phenol-based chemical peeling therapy for cervical and vaginal intraepithelial neoplasia (CIN and VaIN, respectively), with the goal of circumventing obstetric complications associated with surgical treatment and to determine the factors associated with treatment resistance. **Methods:** A total of 483 eligible women diagnosed with CIN, VaIN, or both, participated in this study. Participants underwent phenol-based chemical peeling therapy every 4 weeks until disease clearance. Disease clearance was determined by negative Pap tests for four consecutive weeks or by colposcopy. HPV genotyping was conducted at the onset of the study and after disease clearance in select cases. Our preliminary analysis compared the recurrence and persistence rates between 294 individuals who received phenol-based chemical peeling therapy and 189 untreated patients. **Results:** At 2 years following diagnosis, persistent disease was observed in 18%, 60%, and 88% of untreated patients with CIN1–3, respectively, and <2% of patients with CIN who received phenol-based chemical peeling therapy. Among 483 participants, 10 immune-suppressed patients required multiple treatments to achieve disease clearance, and 7 were diagnosed with cervical cancer. Of the 466 participants, except those with cancer or immune suppression, the number of treatment sessions until CIN/VaIN clearance ranged from 2 to 42 (average: 9.2 sessions). In total, 43 participants (9.2%) underwent surgical treatment. Six patients (1.3%) experienced recurrence of CIN2 or worse, suggesting that treatment failed in 46 patients (9.9%). No obstetrical complications were noted among the 98 pregnancies following this therapy. Factors associated with resistance to this therapy include immune suppression, ages 35–39 years, higher-grade lesions, and multiple HPV-type infections. **Conclusions:** Phenol-based therapy is safe and effective for CINs and VaINs. Women aged < 35 years and with persistent CIN1 or CIN2 with a single HPV-type infection are suitable candidates for phenol-based chemical peeling therapy. However, this therapy requires multiple lengthy sessions.

## 1. Introduction

Cervical cancer is the fourth most prevalent cancer worldwide, and the fourth leading cause of cancer death, accounting for approximately 570,000 new cases and 311,000 deaths in 2018 [1]. The global burden of cervical cancer is unevenly distributed, with >85% of cases occurring in low- and middle-income countries [2]. Over recent decades, trends in cervical cancer incidence and mortality rates have varied among countries. In the United States, there were 13,069 new cervical cancer cases and 4175 deaths in 2015 [1], while in Japan 10,879 new cases and 2921 deaths were reported in 2019 [3]. Among the deaths from cervical cancer in Japan between 2011 and 2019, an average of 340 per year were women aged 20–44 years [3]. Although the incidence and mortality rates of cervical cancer have decreased in many developed countries [4], there has been a significant increase in Japan in the last decade [3].

The World Health Organization recommends surgical resection of premalignant cervical lesions to reduce cervical cancer incidence and mortality in this century [4]. Cold-knife conization and the loop-electric excisional procedure (LEEP) are considered gold standard treatments for high-grade cervical lesions (cervical intraepithelial neoplasia grades 2 and 3; CIN2/3). However, conization, especially with a cold knife, has been associated with an increased risk of preterm delivery and premature rupture of membranes (PROM) post-treatment [5]. The frequency and severity of these adverse sequelae increase with increasing cone depth and repeated treatments. Procedures such as LEEP, conization, and cryotherapy can also potentially induce premature delivery, although the incidence is lower compared to cold-knife conization [6]. To avoid such obstetric complications in young women with CIN2/3 or adenocarcinoma in situ (AIS) planning future pregnancies, less invasive yet effective procedures are desirable.

It is now widely accepted that certain types of human papillomaviruses (HPVs) are the etiological factor for cervical cancer and its precursor lesions [7]. While many women clear HPV infections within a few months, some have persistent infections [8]. This persistent HPV infection is a prerequisite for malignant progression due to the continuous expression of HPV viral oncoproteins E6 and E7, which promote cell proliferation, genomic instability, and the inhibition of host immune responses [9,10]. The most notorious HPV16 infection can be cleared by cell-mediated immunity against E2, E6, and E7 proteins of this type [10,11], and some therapeutic HPV vaccines have been developed to eliminate high-grade squamous intraepithelial lesions (HSILs; worse than CIN2) in some patients [11]. Although certain therapeutic HPV vaccines are well-tolerated, they are unlicensed because of their modest efficacies compared to the use of placebos and surgery (conization) [11]. HPV16 has the ability to evade host immune responses through three mechanisms: non-exposure of HPV antigens during its lifecycle, suppression of certain interferon secretions by HPV16′s E6 and E7 proteins, and acquisition of immune tolerance [10]. Immune tolerance is likely induced following antigen presentation by dendritic cells or macrophages without the stimulation of innate immunity (local inflammation) [12].

In cosmetic dermatology, liquid phenol is utilized to peel the skin, as it penetrates deeply and kills cells [13]. It has been reported that, in 82.6% of patients, HPV-related common warts cleared after 80% phenol therapy, while in 70% they cleared after cryotherapy [14]. More than 80% complete remission is observed in high-grade CINs with a single treatment using trichloroacetic acid (TCA), another skin cosmetic agent [15]. To develop non-invasive treatment methods, we evaluated the efficacy of phenol and TCA therapies in treating cervical or vaginal CIN.

In our preliminary experiment, we applied an 89% phenol solution on the skin of six rats to observe the effects and safety. Within a few days, the skin regions of three rats treated with this solution exhibited erosion, which recovered completely within 2 weeks. All these rats survived and appeared well after the procedure. Another three rats were killed just after the development of erosion to observe damage to the skin. The whole epithelium was damaged, but the bottom of the hair follicles was intact, suggesting that phenol does not penetrate so deeply in the skin. Then we performed chemical peeling on some female participants using 89% phenol as a reagent and found that it was well-tolerated. In this non-randomized prospective study, we investigated the effectiveness and limitations of phenol therapy. This study could provide important information to aid in the development of new therapies, including therapeutic HPV vaccines, for premalignant lesions induced by high-risk (HR) HPV infection.

## 2. Materials and Methods

### 2.1. Subjects and Study Design

This non-randomized prospective study was conducted from September 2009 to April 2018. Initially, 509 women with cervico-vaginal abnormalities were enrolled. The participants were referred to two outpatient clinics of Kanazawa Medical University (Kahoku-gun, Ishikawa) and Yuuai Medical Center (Tomishiro, Okinawa), Japan. Two-thirds of the patients were referred for the treatment or follow-up of low-grade or high-grade cervical or vaginal squamous intraepithelial lesions (LSILs or HSILs), whereas the remaining one-third were referred for HPV testing or colposcopy testing because of positive results in their Pap tests. At enrollment, all participants underwent a histopathological examination guided by colposcopy, and were diagnosed with LSILs or HSILs. For convenience, we used the terminology of intraepithelial neoplasia grades 1–3 (cervico-vaginal intraepithelial neoplasia [CIN]1–3 or vaginal intraepithelial neoplasia [VaIN]1–2) in this study. After obtaining written consent to participate in the study, which was approved by the committee of Kanazawa Medical University (#107), all subjects diagnosed with CIN, VaIN, or both CIN and VaIN, underwent chemical peeling therapy using liquid phenol (phenol therapy). In total, 26 subjects were excluded from the analysis (Figure 1. Therefore, 483 patients were included in this study. The number of treatments required for clearance or termination (cases that needed surgery for invasive cancer or for being resistant to this therapy) was compared among different grades.

Preliminarily we conducted an age-adjusted case-control study with 294 enrolled subjects being treated with phenol therapy and 189 subjects without treatment to investigate the efficacy of chemical peeling therapy with phenol for CINs and VaINs (Figure 3). The controls visited the outpatient clinic of Kanazawa Medical University between November 2001 and September 2020, and received follow-ups for more than 6 months (6–125 months). The final analysis of controls included some cases whose lesions regressed within 12 months or 24 months for each analysis, but the other cases without regression within those periods were excluded for analysis.

The main study included 483 participants with CIN1–3 and VaIN1–2, or both, at entry, but finally they were diagnosed as CIN1 or VaIN1 (*n* = 129), CIN1 and VaIN1 (*n* = 34), CIN2 or VaIN2 (*n* = 149), CIN2 and VaIN1/2 (*n* = 22), CIN3 (*n* = 113), CIN3 and VaIN1/2 (*n* = 19), cancer (*n* = 7), or immune-suppressed CIN or VaIN (IMM-SUP; *n* = 10) (Figure 3). Adenocarcinoma in situ was also present in three patients with CIN3; these AIS that coexisted with CIN3 were resolved during therapy.

### 2.2. Demographic Characteristics of Subjects

At the beginning of the study, information on age, smoking habits (direct or passive), and use of immune-suppressive drugs was obtained from all subjects. Self-reported information on obstetric outcomes post-treatment was gathered from 98 subjects who became pregnant after therapy.

### 2.3. Procedure of the Phenol Therapy

The liquid phenol solution (Maruishi, Seiyaku, Japan) was applied to wide areas of the cervical or vaginal lesions (about 1 cm outside the squamous lesion and about 2–3 cm inside the cervical canal) three to five times using a small cotton tip (3 × 15 mm). In cases where the lesions extended into the cervical canal, the tip was dipped in liquid phenol and inserted into the canal more deeply than usual. However, some women whose lesions were not clearly visible had undergone surgical treatment.

Following the treatment, it was crucial to absorb all of the liquid into the vagina with a cotton ball to prevent phenol solution leakage into the vulva. Treatment was performed at 2-week intervals at the beginning from 2009 to 2011, but it was changed to 4-week intervals later.

### 2.4. Pathological Diagnosis, HPV Testing, and Confirmation of Disease Clearance

Diagnosis was established histologically by two clinical pathologists according to WHO classification (2014). As mentioned above, we used previous terminology such as CIN1–3 and VaIN1–3 in the current study. In repeat colposcopies or following surgical interventions, the final diagnosis was upgraded for 16 cases, including 7 with cervical cancer. Therefore, the final histological diagnosis was used in the present study. Invasive cervical cancer (1A1, 1A2, and 1B1) was identified in seven patients, while the others were diagnosed with additional diseases, primarily VaIN. The lesion size for CIN and VaIN is presented as the number of sections containing the lesion among the 12 clockwise sections of the cervix and vagina. The total number of cervical and vaginal sections containing the lesions were analyzed.

Liquid-based cytology (LBC) samples for Pap tests were collected from some subjects at 4-week intervals, and the other subjects had the test at 12-week intervals. Colposcopy examination was performed to confirm regression of diseases in the latter subjects. Diagnosis was reached through consensus between cytotechnologists and clinical pathologists (blinded to the HPV test results) in the Clinical Pathology Unit at Kanazawa Medical University Hospital or Yuai Medical Center.

A portion of each LBC sample was sent to a commercial laboratory (LSI Medience, Tokyo, Japan) for HPV genotyping. The HPV genotype was determined using a PCR-based test (Genosearch-31; MBL, Nagoya, Japan) [16]. This test can detect 13 HR HPV types (16, 18, 31, 33, 35, 39, 45, 51, 52, 56, 58, 59, and 68), probable high-risk (pHR) types (26, 53, 66, 70, 73, and 82), and low-risk (LR) types (6, 11, 42, 44, 54, 55, 61, 62, 71, 84, 89, and 90). To identify additional pHR-HPV types, such as HPV34, 67, and 69, the uniplex PCR method [17] was used for some CIN or VaIN cases that tested negative with the aforementioned test. In the current study, HPV was grouped as HPV16 and 18 (16/18), higher-grade high-risk type (HGHR; HPV-31, 33, 45, 52, and 58), lower-grade high-risk type (LGHR; HPV-35, 39, 51, 56, 59, and 68), pHR (HPV-26, 30, 34, 53, 66, 67, 69, 70, 73, 82, and 85), and LR type (HPV-6, 11, 40, 42, 44, 54, 61, 62, 71, 72, 74, 81, 84, 89, and 90) or negative (LR/Neg). For cases with multiple HPV types, the highest risk type was considered the primary HPV type responsible for the disease. HPV testing was performed at study entry and at the visit after disease clearance. HPV genotyping was conducted in 477 subjects at entry, and in 327 cases after disease clearance.

Disease clearance was determined based on negative Pap tests for 4 consecutive weeks in some cases or lesion disappearance in the colposcopy in others.

### 2.5. Statistical Analysis

Statistical analysis was performed using the software JMP14 (JMP Statistical Discovery; SAS Institute Inc., Cary, NC, USA) or Prism ver. 9.5.1 (GraphPad Software Inc., San Diego, CA, USA). Treatment times were compared between two groups using the Mann–Whitney U test. *p*-values < 0.05 were considered to indicate statistically significant differences. Logistic regression analysis was also conducted to identify independent risk factors for treatment resistance. The cut-off number of treatments indicating treatment resistance was calculated as 13 treatments, as this exceeded the 75th percentile of number of treatments until disease clearance among all cases.

## 3. Results

### 3.1. Clinical Course of Cervico-Vaginal Abnormal Lesions with Phenol Therapy

Two representative cases of use of phenol therapy are illustrated in Figure 2. Case 1 involved a 24-year-old woman with HPV16-positive CIN3, and Case 2 involved a 20-year-old woman with multiple CIN3, VaIN2, and cervicovaginal condyloma lesions positive for HPV6, 16, and 66. In these instances, 17 and 22 phenol therapy sessions were conducted, respectively, until complete lesion and HPV clearance. The former case involved a passive smoker. Some patients experienced dizziness, palpitations, and lower abdominal discomfort during or after phenol application. While these symptoms usually disappeared within 10 min after treatment, in some cases they persisted for a day. Lidocaine gel was used in some cases before treatment to alleviate local pain or discomfort. Two patients chose to discontinue the treatment due to skin rashes (possibly allergic reactions), while thirteen patients dropped out of the study for unknown reasons (Figure 3).

We compared the rates of disease regression and persistence between participants treated with phenol therapy and untreated controls (Figure 2). The 12-month disease regression rates for CIN1–3 were 44%, 11%, and 0% for controls, respectively, and 86%, 76%, and 59% for the treated patients, respectively. The 24-month disease regression rates for CIN1–3 were 65%, 22%, and 0% for controls, respectively, and 99%, 98%, and 100% for the treated patients, respectively. These results suggest that the 12- and 24-month disease regression rates were higher in treated subjects than controls (Fisher’s test, *p* < 0.0001). Conversely, the controls had significantly higher 2-year disease persistence rates for CIN1–3 (18%, 60%, and 88%, respectively) compared to the treated patients (<2%).

The current study analyzed 483 eligible subjects at the time of enrollment in this study (Figure 3), including those with CIN1 and VaIN1 (*n* = 163), CIN2 and VaIN1/2 (*n* = 171), CIN3 and VaIN1/2 (*n* = 132), cancer (*n* = 7), and immune-suppressed CIN or VaIN (IMM-SUP; *n* = 10). The IMM-SUP group contained cases of CIN1 or VaIN (*n* = 2), CIN1 and VaIN1 (*n* = 2), CIN2 (*n* = 4), and CIN3 (*n* = 2). Cases with IMM-SUP needed additional treatment sessions, with an average number of treatment sessions of 20.9 (95% CI: 11.6–30.2) compared to 9.1 (95% CI: 8.5–9.7) for the remaining cases. Therefore, these cases were excluded from further analysis due to the substantial impact of immune suppression on therapy outcomes (Figure 4A).

Cervical cancer IA1 was identified in one case after a hysterectomy, while the remaining six cancer cases were diagnosed prior to surgery. Most of these cases were resistant to phenol therapy. The cancer diagnoses included stages 1A1 (*n* = 3), 1A2 (*n* = 2), and 1B1 (*n* = 2), with four cases of squamous cell carcinoma (SCC) and three of adenocarcinoma (ADC). One 1A1 case underwent conization, another underwent a simple hysterectomy, and the rest received radical hysterectomies with pelvic lymph node resection. All of these patients have remained disease-free as of September 2023.

Among 466 cases, excluding those with cancer or in the IMM-SUP group, the number of treatment sessions until disease clearance ranged from 2 to 42 (average = 9.2, 95% CI: 8.56–9.77). Higher-grade CINs required a greater number of treatment sessions than lower-grade ones, with average numbers of session of 6.1, 9.9, and 11.9 for CIN1 and VaIN1, or both, CIN2 and VaIN1/2, or both, and CIN3 and VaIN1/2, or both, respectively (Figure 4A). CIN2 or VaIN2 and CIN3 needed a greater number of treatment sessions than CIN1 or VaIN1, whereas CIN3 needed a greater number of sessions than CIN2 or VaIN2. Conversely, the average numbers of treatment sessions until disease clearance for both CIN1 and VaIN1, CIN2 and VaIN1/2, and CIN3 and VaIN1/2 were 11.1 (95% CI: 9.13–13.0), 14.3 (95% CI: 9.30–19.2), and 14.1 (95% CI: 10.5–17.6), respectively, indicating that there was no difference among the grades of CINs and VaINs (Figure 4B). Cases with coexisting CIN and VaIN required a greater number of treatments than those with CIN1 or VaIN1 and CIN2 or VaIN1/2. However, no difference was observed between CIN3 or VaIN3, and CIN3 and VaIN1/2 (Figure 4B).

The prevalence rates of HPV 16 or 18 (16/18), HGHR, LGHR, pHR, and LR/Neg were 29.8% (*n* = 137), 37.8% (*n* = 174), 10.7% (*n* = 49), 6.3% (*n* = 29), and 15.4% (*n* = 71), respectively. HPV tests were not performed in six cases. HPV16/18, HGHR, and pHR required a greater number of treatment sessions than LR/Neg. Furthermore, HPV16/18 needed a greater number of treatment sessions than HGHR and LGHR. However, no difference was observed in the treatment time between HPV16/18 and pHR or between LGHR and LR/Neg (Figure 5A). The higher-risk group (HPV16/18, HGHR, and pHR) required a greater number of treatment sessions than the lower-risk group (LGHR, LR, and HPV-negative) (Mann–Whitney U test; *p* < 0.0001). Multiple HPV-type infections needed a greater number of treatment sessions than single HPV-type infection or LR/Neg (Figure 5B).

### 3.2. Management of Therapy-Resistant and Recurrent Cases

Among the 466 cases, 43 (9.3%) underwent surgical treatment based on the clinicians’ or patients’ decisions. Some cases were diagnosed with CIN1 or VaIN1 (*n* = 3), CIN2 or VaIN2 (*n* = 11), CIN2 and VaIN1 (*n* = 2), CIN3 (*n* = 26), and CIN3 and VaIN (*n* = 1).

Seven cases with CIN2 and VaIN1/2, or both, (1.5%) and ten with CIN3 and VaIN1/2, or both, (2.2%) developed recurrence 6 months following disease clearance. In total, 17 cases with CIN2 or CIN3 and VaIN1/2, or both, (3.6%) developed disease recurrence, of which 6 (1.3%) developed CIN2 or worse. Ten recurrent cases were completely cured by additional phenol therapy, three underwent LEEP surgery, and four remained classified as LSIL or ASCUS at the conclusion of the study. The latter cases were treated with TCA therapy (a chemical peeling therapy that we implemented in May 2018). In total, 46 patients needed surgery or developed recurrence of CIN2 or worse, suggesting that the treatment failure rate was 9.9%.

### 3.3. Obstetric Outcomes in Subsequent Pregnancies

Ninety-eight subjects reported their obstetric experiences post-therapy to clinicians. Of these, 97 subjects (99%) had full-term deliveries. One case resulted in premature delivery at 35 weeks without any obstetrical complications. Four subjects underwent LEEP after phenol therapy but still had normal deliveries.

### 3.4. Factors Conferring Resistance to Phenol Therapy

We sought to determine the factors associated with resistance to this treatment among 466 subjects. A univariate analysis revealed that factors such as being 35–39 years of age, having higher-grade lesions, higher-grade HR (HPV16/18, HGHR, and pHR), multiple HPV-type infections, wide lesions, and passive smoking were significantly associated with resistance (Table 1).

Multivariate analysis indicated that being 35–39 years of age, having higher-grade lesions, and having multiple HPV-type infections were associated with treatment resistance (Table 1). Conversely, HPV type, lesion size, and smoking were not significant in the multivariate analysis.

## 4. Discussion

This case-control study had certain limitations due to its non-randomized interventional design but revealed that persistent disease at 2 years was observed in 18%, 60%, and 88% of untreated controls with CIN1–3, respectively, and <2% of patients treated with chemical peeling therapy with phenol. This suggests that the chemical peeling method is effective to clear CIN1–3. On the other hand, the present study demonstrated that additional surgical treatment was needed in 9.2% (43/466). The recurrence for CIN2 or worse was observed in 6 cases (1.3%; 6/466), and, therefore, treatment failure was 9.9% (46/466). Among the cases with CIN2/3 or VaIN2 or cancer, additional surgical treatment was needed in 16.1% (50/310), and treatment failure was 18.1% (56/310). A meta-analysis of 96 studies on the surgical treatment of precancerous cervical lesions (CIN2 and CIN3) reported a recurrence rate of 6.6% (95% CI: 4.9–8.4) [18]. The rates were found to be heterogeneous (range: 1.4–18.4) and varied by treatment procedure, being 2% for cold-knife conization and laser conization and 7% for large loop excision of the transformation zone [18]. A multicenter study in Japan with 14,832 cases (median age = 37, range: 16–88 years) [19], had a recurrence rate of 6.1%, more advanced diseases were found in 10.5%, and additional treatments were needed in 14.7%. Long-term follow-up studies demonstrated that the 5- and 10-year risks of developing post-treatment CIN2 and CIN3 were 16.5%, 8.6%, 18.3%, and 9.2%, respectively [20]. Most of our patients were followed for 5–14 years, suggesting that the recurrence rate may be lower, but the treatment failure rate is similar to that of surgical therapies.

Guidelines by several groups, including the International Society for the Study of Vulvovaginal Disease, do not recommend TCA therapy (a kind of chemical peeling therapy) for vulvo-vaginal diseases [21]. In the present study, the average numbers of treatment sessions required for disease clearance were 4.8 and 6.7 for VaIN1 and VaIN2, respectively, which did not differ from those needed for CIN1 and CIN2 (*p* > 0.1, Mann–Whitney U test). The clearance rates were 100% (13/13) and 83% (5/6) for cases with VaIN1 and VaIN2, respectively, suggesting that this treatment method is effective for VaIN1/2.

Changes in the average age at first marriage and remarriage for women from 1989 to 2018 (25.8–29.4 and 36.5–40.4 years, respectively) [22] suggest that many women who wish to have children are at high risk of developing premalignant cervical lesions. Although cervical conization with a cold knife and LEEP are the gold standard treatments for high-grade cervical lesions (CIN2/3) [4], such surgical treatments have an obstetric disadvantage. Relative risks (RRs) for premature delivery after conization with a cold knife, laser conization, and large loop excision of the transformation zone (LLETZ) are 2.7, 2.11, and 1.57, respectively [5]. This risk also exists for LEEP and cryotherapy [6]. Phenol therapy showed almost no adverse obstetric outcomes in 98 pregnant cases, including four women who received LEEP treatment after phenol therapy.

The key disadvantage of phenol therapy is the need for multiple treatment sessions (range: 2–42; average = 9.2, 95% CI: 8.56–9.77) until CIN and VaIN clearance. It may be necessary to carefully select suitable cases for this therapy and recommend alternative treatments for those who are likely to resist this therapy. Thus, we examined factors contributing to resistance to this therapy. We found that the risk of treatment resistance was increased for those aged 35–39 years, those with higher-grade lesions, and those with multiple HPV-type infections. The optimal candidates for this therapy include patients with future plans for pregnancy, those who are immunocompromised, those aged < 35 years, those with persistent CIN1 or CIN2, and those with a single HPV-type infection.

The group with the highest prevalence of cervical cancer in Japan is women aged 30–49 years, while the group with the highest prevalence of premalignant lesions (HSIL, CIN2/3) is women aged 25–34 years [3]. The peak age of the incidence of cervical cancer from 2011–2015 in Japan was 40–44 years. This suggests that most women aged 35–39 years with CIN2/3 are likely to be at the highest risk for progressing to invasive cancer within 5 years. Although HPV infection type (such as 16 and 18) was not associated with the risk, multiple HPV-type infections confer resistance against the phenol therapy. Several studies have suggested that a multiplicity of HPV types is associated with persistent HPV infection [23] and the presence of HSILs [23,24,25]. In particular, infection with HPV16 and other HPV types has strongest association [24,25].

In the present study, immune suppression was associated with a large number of treatment sessions, suggesting that host immunity is the most important factor contributing to resistance to this therapy. Many studies have already suggested that the host immune response is important for persistent HPV infection and cervical cancer development [7,8,9]. Other studies have also suggested that HIV-positive individuals are at higher risk for cervical cancer [26,27] and oropharyngeal cancer [28].

In the natural history of HPV infections leading to cancer, smoking, hormonal exposure, and HIV are additional factors that increase the risk of progression to cancer [29]. Some studies have suggested that not only active smokers, but also passive [30,31] and tertiary smokers [31], are at high risk for cervical cancer. The present study also demonstrated that passive smoking is associated with treatment resistance, although the association was not significant in the multivariate analysis. A weak association of smoking with treatment resistance is likely to emerge in the present study, as many smokers had quit smoking or avoided passive smoking due to awareness provided by practitioners that smoking promotes cervical cancer development. Some studies show that smoking cessation after a cancer diagnosis enhances the therapeutic response [32]. Furthermore, smoking affects the host immune response [33] against persistent HPV infection, especially in high-grade CINs. Immune tolerance to some HPV antigens or an immune-suppressive state induced by immune checkpoint inhibitors may be present in higher-grade lesions infected with HR HPV types [10,12]. Some therapeutic HPV vaccines have been successful in treating HPV-related premalignant lesions [11]. However, complete clearance by such therapy has not yet been established.

To increase the efficacy of any type of treatment for cervical premalignant lesions or cancer, we need to better understand the mechanisms of host defense against persistent HPV infections and certain driver gene mutations in host cells progressing to cancer. The vaginal microbiota [34] and immune microenvironment [35] may also be important for a comprehensive understanding of host defense systems against HPV-related cancer.

## 5. Conclusions

Chemical peeling therapy using phenol is as effective and safe as conization. The main limitations of this treatment method are the need for multiple, lengthy treatment sessions for some patients. Modifications to the treatment are needed to increase the applicability of chemical peeling therapy in clinical settings. The optimal candidates for this therapy are likely to be patients with future plans for pregnancy, who are immunocompromised, who are <35 years, who have persistent CIN1 or CIN2, and who have a single HPV-type infection.

## Figures and Tables

**Figure 1 viruses-15-02219-f001:**
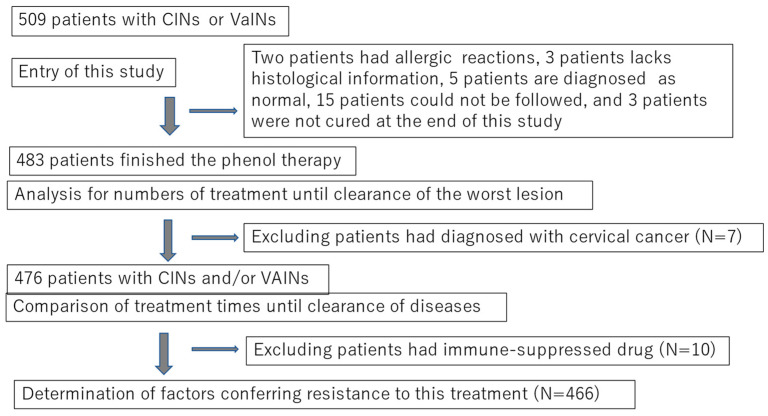
**Design of the main study.**

**Figure 2 viruses-15-02219-f002:**
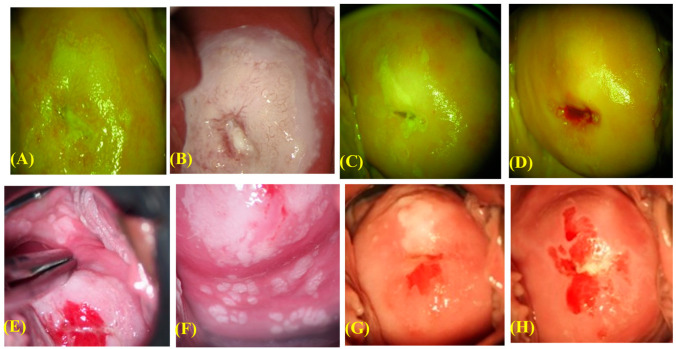
Two instances of CIN3, with or without the presence of VaIN2. Case-1; 24 years old woman having CIN3, HPV16-positive, Passive smoker. (**A**) Colposcopy finding after treating with 3% of acetic acid showing white epithelium and mosaic pattern was seen in 10 out of 12 sections of the cervix. (**B**) After application of liquid phenol, (**C**) After 11 times of treatment for 9 months. (**D**) After 16 times of treatment for 12 months. This case was cured with 18 times of treatment for 13 months. Case-2; 20 years old having CIN3 + VaIN2, positive with HPV16 + HPV66 + HPV6, Passive smoker. (**E**) Colposcopic finding after applying with 3% of acetic acid. White epithelia were seen in 12 out of 12 sections of the cervix and 6 of 12 sections of the vagina. (**F**) After applying with a liquid phenol, (**G**) After 8 times of treatment. (**H**) After 20 times of treatment. This case was cured with 22 times of treatment for 11 months. Both cases were treated by this therapy at 2 weeks intervals.

**Figure 3 viruses-15-02219-f003:**
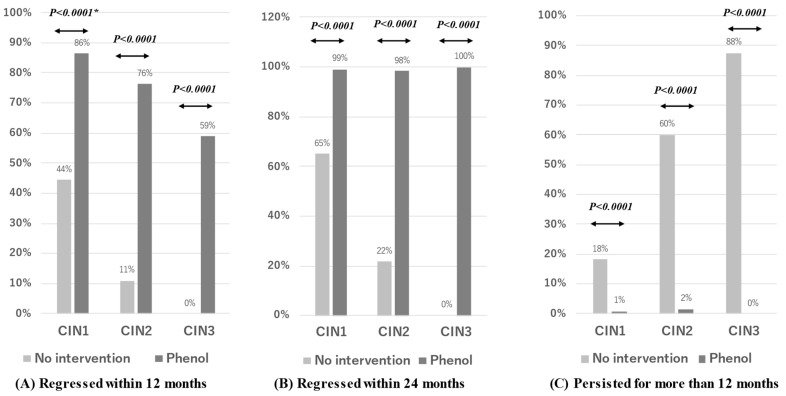
A comparison for regression and persistence between subjects undertaking phenol therapy and those without any treatment.Phenol; Subjects with phenol therapy; CIN1: N = 110, CIN2: N = 123, CIN3: N = 61. No intervention; Controls without any treatment; CIN1: N = 126, CIN2: N = 55, CIN3: N = 8. *; Probability in comparison of regressed or persisted rates in Phenol and No intervention cases was calculated in the Mann-Whitney’s test.

**Figure 4 viruses-15-02219-f004:**
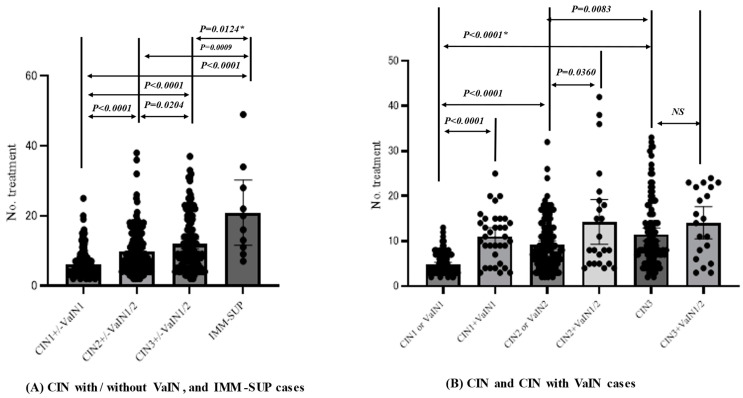
Differences in the number of treatments until clearance between different grades of CIN, and an immune-suppressed status. *; Probability in comparison of numbers of treatment until disease clearance among disease groups was calculated in the Mann-Whitne’s test. NS: Not significant.

**Figure 5 viruses-15-02219-f005:**
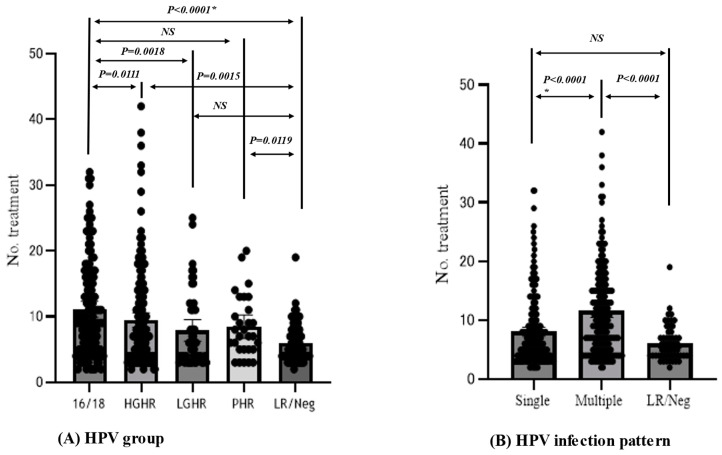
**Difference in the number of treatment until clearance between HPV type groups and different infection patterns.** (**A**) HPV group, HPV16/18; HPV16 or HPV18, HGHR; higher grade high-risk types: HPV31, 33, 45, 52, 58, LGHR; lower grade high-risk types: HPV39, 51, 56, 59, 68, PHR; probable high-risk types; HPV26, 53, 66, 67, 70, 73, 82. LR/ Neg; Low-risk types: HPV6, 11, 42, 44, 54, 55, 61, 62, 71, 84, 89, and 90 or negative for HPV. (**B**) Single HPV type and multiple HPV type infection. *; Probability in comparison of numbers of treatment until disease clearance among disease groups was calculated in the Mann-Whitne’s test. NS: Not significant.

**Table 1 viruses-15-02219-t001:** The factors that contribute to resistance to phenol therapy.

	Monovariate Analysis			Multivariate Analysis		
Variable Factors	ORs	95%CI	Probability	ORs	95%CI	Probability
**Age group**			*NS*			*p = 0.0344 (Wald test)*
35–39 yrs vs. 16–29 yrs	1.0	0.54–1.91	*NS*	3.6	1.37–9.26	*p = 0.0090*
35–39 yrs vs. 30–34 yrs	1.7	0.89–3.36	*NS*	4.5	165–12.4	*p = 0.0034*
35–39 yrs vs. 40–44 yrs	1.6	0.82–3.06	*NS*	3.4	1.29–9.02	*p = 0.0134*
35–39 yrs vs. over 45 yrs	1.4	0.71–2.77	*NS*	2.8	1.04–7.79	*p = 0.0415*
**Grade of lesion**						*p < 0.0001(Wald test)*
CIN2 or/andVAIN1/2 vs. CIN1 or/and VAIN1	3.5	1.92–6.62	*p < 0.0001*	3.4	1.58–7.31	*p = 0.0017*
CIN3 or/and VAIN1/2 vs. CIN1 or/and VAIN1	5.3	2.86–10.1	*p < 0.0001*	7.8	3.29–18.6	*p < 0.0001*
CIN3 or/and VAIN1/2 vs. CIN2 or/andVAIN1/2	1.5	0.93–2.46	*NS*	2.3	1.14–4.64	*p = 0.0198*
**HPV group (a)**						*NS (Wald test)*
HGHR/PHR vs. LGHR/ LR/ Neg	4.6	2.44–9.81	*p < 0.0001*	1.6	0.64–3.77	*NS*
**Multiple or Single type HPV infection**						*p = 0.0008 (Wald test)*
Multiple vs. Negative	3.6	2.31–5.63	*p < 0.0001*	3.1		*1.59–5.85*
**Lesion area size (b)**						*NS (Wald test)*
Over 4 sections vs. 1–3 sections of Cx/Vag	2.1	1.14–3.94	*p = 0.0162*	1.3	0.63–2.78	*NS*
**Active/passive smoking**						*NS*
Direct/passive smoking vs. no smoking	1.9	1.19–2.92	*p = 0.0063*	1.4	0.76–2.62	*NS*

(a) HGHR/PHR; HPV16, 18, 31, 33, 45, 52, 58/HPV26, 30, 34, 53, 66, 67, 69, 70, 73, 82, 85, LGHR/LR/Neg; HPV35, 39, 51, 56, 59, 68/HPV6, 11, 40, 42, 44, 54, 55, 61, 62, 71, 74, 81, 84, 89, 90, and negative with HPV. (b) lesion size was defined as calculated by adding x sections of 12 clockwise sections of the cervix and y sections of that of the vagina. NS: Not significant.

## Data Availability

The data presented in this study are available on request from the corresponding author. The data are not publicly available to ensure the protection of privacy.

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
