# Peer review of "Chemical Peeling Therapy Using Phenol for the Cervico-Vaginal Intraepithelial Neoplasia"

_viruses, 2023, doi:10.3390/v15112219_

Round 1

Reviewer 1 Report

Comments and Suggestions for Authors

In this manuscript, the authors described "Phenol-based Chemical Peeling Therapy" as a treatment for HPV-infected lesions or pre-cancer lesions. As there is no specific treatment for HPV infection, current treatments are basically based on physical removal/destruction of the area of infected cells. To achieve the goal, of "Elimination of cervical cancer", it is the key, to how to treat the pre-cancer lesions which are found by cervical cancer screening.

With this context, I agree that this kind of study should be publicised in general. However, the manuscript is not presented in a proper/sufficient way by the current format, and a major revision/re-write is required to publish in "Viruses"

Major points.

1. The aim/focus of the manuscript is not clearly presented both in the title/abstract and the main text. It could be the efficacy, safety, determination factors of the treatment or obstetric complications following the treatment. In the abstract, "to validate the use of...." it sounds too vague, and the ambiguity results in the study focus. According to the study focus the authors presented "determination of factors conferring resistance to Phenol-based Chemical Peeling Therapy" seems the main focus. Then no conclusion in the abstract.

2. With point "1", the authors must pay more attention to the limitations of non-randomised intervention studies. Nonrandomised studies may give seriously misleading results when treated and control groups (this study has no control group) appear similar in key prognostic factors. Standard methods of case-mix adjustment do not guarantee the removal of bias. Residual confounding may be high even when good prognostic data are available (please see Health Technology Assessment 2003; Vol. 7: No.27). The authors do not provide sufficient discussions in this aspect in this manuscript, generally overstating the interpretation which can be led by the data presented.

2.1 In the first paragraph of the discussion, the authors discussed the efficacy of Phenol-based Chemical Peeling Therapy. It seems the main conclusion in this manuscript, but it is not clearly presented what are the 320 patients even in the study design.

The determination factors, "being 35–39 years of age, having high-grade lesions (CIN2 or worse), high-risk or probable high-risk HPV. type-positive, immune suppression, and smoking", are quite similar as the determination factors of spontaneous regression of CIN without any interventions, suggesting the efficacy of the therapy may have "residual confounding". Especially, the number is based on patients after the removal of 52 patients who required surgical intervention and might have lesions with higher recurrence rates, as such I am not convinced that the rate of failure is very low (1.9%). And the number is overstating and misleading. The demographics of patients in each study can explain the differences, at least to some extent, in figures.  Please discuss this point.

3 With the third and fourth paragraph in discussion, the authors are claiming "It may be necessary to carefully select suitable cases for this therapy and recommend alternative treatments for those who are likely to resist this therapy." If so, based on the results presented, who is supposed to have the peeling therapy?  It can be read in the manuscript that 35-39 years of age, having CIN2+ lesion, with high-risk HPV, who has the highest risk for developing invasive cervical cancer in the next 5 years, should avoid this treatment. This point must be discussed clearly in the revised manuscript.

3.1 The authors claim "However, immune suppression was a stronger predictor than any grade of CINs or VaINs, suggesting that immune suppression is the most important factor contributing to resistance to this therapy". It is not shown in the manuscript (Table 1) and, based on the study design, patients with immune suppression seem to be removed in this study. It sounds confusing.

3.2 Taking "3" and "3.1" into account, the factors conferring resistance to the therapy might be reflecting "residual confounding" which affects spontaneous regression without any intervention. Please discuss this point.

4 The definition of success/end of treatment and how it was determined must be presented.

5 Seven patients were diagnosed with invasive cancer during the treatment. It is more than 1% of the patients enrolled with cervical abnormality (ASC-US +). It sounds like a key disadvantage of this therapy. Please discuss this.

6 The number of treatments rather than the time required for clearance is shown. The observational periods must affect the rate of clearance. Or does the number of treatment reflects roughly the observational periods? Please clarify it

Minor points

1 Line 63 ref 8. This manuscript is not about clearance.

2 Line 68 ref 10. Based on the reference, E7 is not targeted by host immunity?

3 Line 68 I agree the proof of concept of therapeutic vaccine is demonstrated but still under development.

4 The mode of action of Phenol therapy on HPV lesions can be explained in the Introduction a bit in detail.

5 In M and M. All patients with ASC-US+ and HPV+ had CIN1+. Is it right? If so, it sounds unusual. No cases without CIN lesions under colposcopy in this study?

6 Line 171-172. The changes in diagnosis are down/upgraded, but it is not described clearly enough. Many cases would be downgraded during the treatment anyway, and the initial diagnosis won't be affected. What does this mean?

7 Is it the best way to show the number of treatments in the median and 95% CI? Line 188-

Author Response

#Reviewer-1
Q1. The aim/focus of the manuscript is not clearly presented both in the title/abstract and 
the main text. It could be the efficacy, safety, determination factors of the treatment or 
obstetric complications following the treatment. In the abstract, "to validate the use of...." 
it sounds too vague, and the ambiguity results in the study focus. According to the study 
focus the authors presented "determination of factors conferring resistance to Phenolbased Chemical Peeling Therapy" seems the main focus. Then no conclusion in the 
abstract.
A1. We agree with this comment. We put the comment in objective of the abstract as 
follows; This study aimed to validate the use of liquid phenol-based chemical peeling 
therapy for cervical and vaginal intraepithelial neoplasia (CIN and VaIN, respectively)
with the goal of circumventing obstetric complications associated with surgical treatment
and to determine the factors associated with treatment resistance. (the first paragraph, 
Page 2)
Q2. With point "1", the authors must pay more attention to the limitations of nonrandomised intervention studies. Nonrandomised studies may give seriously misleading 
results when treated and control groups (this study has no control group) appear similar 
in key prognostic factors. Standard methods of case-mix adjustment do not guarantee the 
removal of bias. Residual confounding may be high even when good prognostic data are 
available (please see Health Technology Assessment 2003; Vol. 7: No.27). The authors 
do not provide sufficient discussions in this aspect in this manuscript, generally 
overstating the interpretation which can be led by the data presented.
A2. We agree with this comment. We extensively revised this manuscript. First we added 
an age-adjusted case-control study, and new comments were added as follows; 
#1. Our preliminary analysis compared the recurrence and persistence rates between 294 
individuals who received phenol-based chemical peeling therapy and 189 untreated 
patients. (The second paragraph, Page 2)
#2. At 2 years following diagnosis, persistent disease was observed in 18%, 60%, and 
88% of untreated patients with CIN1–3, respectively, and <2% of patients with CIN who 
received phenol-based chemical peeling therapy. (the third paragraph, Page 2)
#3. Preliminarily we conducted a age-adjusted case-control study enrolled 294 subjects 
being treated with phenol therapy and 189 subjects followed without treatment to 
investigate the efficacy of chemical peeling therapy with phenol for CINs and VaINs 
(Fig. 2). The controls visited the outpatient clinic of Kanazawa Medical University 
between November 2001 and September 2020, and were followed up for more than 6 
months (6–125 months). The final analysis of controls included some cases whose 
lesions regressed within 12 months or 24 months for each analysis, but the other cases 
without regression within those periods were excluded for analysis. (The second 
paragraph, Page 6 in Materials and methods)
#4, We compared the rates of disease regression and persistence between participants 
treated with phenol therapy and untreated controls (Fig. 2). The 12-month disease 
regression rates for CIN1–3 were 44%, 11%, and 0% for controls, respectively, and 86%, 
76%, and 59% for the treated patients, respectively. The 24-month disease regression rates 
for CIN1–3 were 65%, 22%, and 0% for controls, respectively, and 99%, 98%, and 100% 
for the treated patients, respectively. These results suggest that the 12- and 24-month 
disease regression rates were higher in treated subjects than controls (Fisher’s test, p < 
0.0001). Conversely, the controls had significantly higher 2-year disease persistence rates 
for CIN1–3 (18%, 60%, and 88%, respectively) compared to the treated patients (< 2%).
(The second paragraph, Page 10)
Q2.1 In the first paragraph of the discussion, the authors discussed the efficacy of Phenolbased Chemical Peeling Therapy. It seems the main conclusion in this manuscript, but it 
is not clearly presented what are the 320 patients even in the study design.
The determination factors, "being 35–39 years of age, having high-grade lesions (CIN2 
or worse), high-risk or probable high-risk HPV. type-positive, immune suppression, and 
smoking", are quite similar as the determination factors of spontaneous regression of CIN 
without any interventions, suggesting the efficacy of the therapy may have "residual 
confounding". Especially, the number is based on patients after the removal of 52 patients 
who required surgical intervention and might have lesions with higher recurrence rates, 
as such I am not convinced that the rate of failure is very low (1.9%). And the number is 
overstating and misleading. The demographics of patients in each study can explain the 
differences, at least to some extent, in figures. Please discuss this point.
A2-1. 
This is the most important question. We added some comments to show a limitation 
of this study in the abstract, result and discussion as follows; 
#1. In total 43 participants (9.2%) underwent surgical treatment. Six patients (1.3%) 
experienced recurrence of CIN2 or worse, suggesting that treatment failed in 46 patients 
(9.9%). (from the last paragraph, page 2 to the first one , Page 3)
#2. After excluding IMM-SUP cases, cervical cancer were found in 7 cases, 
demonstrating that more advance disease was found in 1.4% (7/473), and surgical 
treatment needed 10.6% (50/473). (from the last paragraph, page 10 to the first one, Page 
11 in the result)
#3. On the other hand, the present study demonstrated that additional surgical treatment 
needed 9.2% (43/ 466). The recurrence for CIN2 or worse was observed in 6 cases (1.3%; 
6/ 466), and, therefore, treatment failure was 9.9% (46/ 466). If cancer cases were 
included, additional treatment needed in 10.6% (50/473), and treatment failure was seen 
in 11.8% (56/473). When limited to the cases with CIN2/3/VaIN2 or worse, additional 
surgical treatment needed 16.1% (50/310), and treatment failure was 18.1% (56/310).
(The last paragraph, Page 13 in the discussin)
Q3 With the third and fourth paragraph in discussion, the authors are claiming "It may be 
necessary to carefully select suitable cases for this therapy and recommend alternative 
treatments for those who are likely to resist this therapy." If so, based on the results 
presented, who is supposed to have the peeling therapy? It can be read in the manuscript 
that 35-39 years of age, having CIN2+ lesion, with high-risk HPV, who has the highest 
risk for developing invasive cervical cancer in the next 5 years, should avoid this 
treatment. This point must be discussed clearly in the revised manuscript.(from the third 
paragraph, Page 2 to the first one, Page 3 in the abstract)
A3. We agree with this comment, and add new comment in the abstract, the discussion 
and the conclusion as follows; 
#1. Women of age < 35 years and with persistent CIN1 or CIN2 with single HPV type 
infection are suitable for phenol-based chemical peeling therapy. (The second paragraph, 
Page 3 in the abstract)
#2. We found that the risk of treatment resistance was increased by age of 35–39 years, 
having higher-grade lesions, and multiple HPV type infections. The optimal candidates 
for this therapy include patients with future plans for pregnancy, who are 
immunocompromised, age < 35 years, persistent CIN1 or CIN2, and single HPV type 
infection. (The second paragraph, Page 15 in the discussion).
#3. The optimal candidates for this therapy are likely to be patients with future plans for 
pregnancy, who are immunocompromised, age < 35 years, persistent CIN1 or CIN2, and 
single HPV type infection. (The first paragraph, Page 17 in the conclusion)
3.1 The authors claim "However, immune suppression was a stronger predictor than any 
grade of CINs or VaINs, suggesting that immune suppression is the most important factor 
contributing to resistance to this therapy". It is not shown in the manuscript (Table 1) and, 
based on the study design, patients with immune suppression seem to be removed in this 
study. It sounds confusing.
A3.1.; According to this suggestion, we extensively revised the manuscript, Figures and 
Table-1 according to this precious comment, However, we confess that some results are 
changed by this modification, although the result seems to become more clear than before.
HPV type group and smoking became not significant as the factors conferring resistance 
to this therapy in the multivariate analysis. a new comment was added as follows;
Cases with IMM-SUP needed additional treatment sessions, with an average number of 
treatment sessions of 20.9 (95% CI: 11.6–30.2) compared to 9.1 (95% CI: 8.5–9.7) for 
the remaining cases. Therefore, these cases were excluded from further analysis due to 
the substantial impact of immune suppression on therapy outcomes (Fig. 4A). (The last
paragraph, Page 10) .
Q3.2 Taking "3" and "3.1" into account, the factors conferring resistance to the therapy 
might be reflecting "residual confounding" which affects spontaneous regression without 
any intervention. Please discuss this point.
A3.2. In order to show regression of diseases by chemical peeling method, a age-adjusted 
case-control analysis was performed in this study as mentioned above. This result showed 
that regression rate by this therapy was apparently higher than that by spontaneous 
regression in controls without treatment. This clearly indicates that the phenol therapy act 
on regressing CIN1-3 lesions. 
Q4 The definition of success/end of treatment and how it was determined must be 
presented.
A4. According to comments by this referee, we added some comments to show rates of 
additional surgical treatment, recurrence and treatment failure (surgery and recurrence)
as follows; 
#1. In total 43 participants (9.2%) underwent surgical treatment. Six patients (1.3%) 
experienced recurrence of CIN2 or worse, suggesting that treatment failed in 46 patients 
(9.9%). (from the last paragraph in page 2 to the first line in Page 3)
#2. Seven cases with CIN2 and/or VaIN1/2 (1.5%) and ten with CIN3 and/or VaIN1/2 
(2.2%) developed recurrence 6 months following disease clearance. Seventeen cases with 
CIN2 or CIN3 and/or VaIN1/2 (3.6%) developed disease recurrence, of which 6 (1.3%) 
developed CIN2 or worse.. (the third paragraph in Page 12)
#3. On the other hand, the present study demonstrated that additional surgical treatment 
needed 9.2% (43/ 466). The recurrence for CIN2 or worse was observed in 6 cases (1.3%; 
6/ 466), and, therefore, treatment failure was 9.9% (46/ 466). Among the cases with 
CIN2/3 or VaIN2 or cancer, additional surgical treatment needed 16.1% (50/310), and 
treatment failure was 18.1% (56/310). (The last paragraph in Page 13 in Discussion)
Q5 Seven patients were diagnosed with invasive cancer during the treatment. It is more 
than 1% of the patients enrolled with cervical abnormality (ASC-US +). It sounds like a 
key disadvantage of this therapy. Please discuss this.
A5. The present study is not the population-based study for cervical cancer screening. 
Many cancer cases were found from two third of LSIL or HSIL cases who had been 
already diagnosed at entry. Therefore, 2.3% of cancer among 310 CIN2/3 cases is not so 
many when compared with some reports for conization. To avoid confusion, we added 
the following comments;
#1; Two-third of the patients were referred for the treatment or follow-up of low-gradeor high-grade cervical or vaginal squamous intraepithelial lesions (LSILs or HSILs),
whereas the remaining were referred for HPV testing or colposcopy test because of 
cervical cell abnormalities in cervical cancer screening. (The first paragraph in Page 6
in Materials and Methods)
#2; A multicenter study in Japan with 14,832 cases (median age = 37, range: 16–88 years) 
(19), had a recurrence rate of 6.1%, more advance diseases are found in 10.5%, and 
additional treatment are needed in 14.7%. (The last paragraph, Page 13 in the discussion)
Q6 The number of treatments rather than the time required for clearance is shown. The 
observational periods must affect the rate of clearance. Or does the number of treatment 
reflects roughly the observational periods? Please clarify it
A6. This is reasonable question. We adopted numbers of treatment in this study, since 
numbers of treatments basically represented terms for treatment until clearance. In 
addition, data of terms for treatment until clearance lacked in some cases, and the interval 
of treatment changed in the other cases. 
Minor points
1 Line 63 ref 8. This manuscript is not about clearance.
A1. Reference was replaced to correct one.
Q2 Line 68 ref 10. Based on the reference, E7 is not targeted by host immunity?
A2. The author of ref.10 may believe that HPV16 E7 is not so much effective for 
cytotoxic T lymphocyte activity, because of a lack of epidemiological data regarding E7 
responses. But we did not cite this reference in the revised manuscript.
Q3 Line 68 I agree the proof of concept of therapeutic vaccine is demonstrated but still 
under development.
A3. We added a comment; 
Although certain therapeutic HPV vaccines are well-tolerated, they are unlicensed 
because of their modest efficacies compared to placebo and surgery (conization) (11). 
(The last paragraph, Page 4)
Q4 The mode of action of Phenol therapy on HPV lesions can be explained in the 
Introduction a bit in detail.
A4. We added some comments in the introduction; 
In preliminary experiment, we applied 89% phenol solution on the skin of six rats to 
observe an effect and safety. Within a few days, the skin region of three rats treated with 
this solution exhibited erosion, which recovered completely within 2 weeks. All these rats 
survived and appeared well after the procedure. Another three rats were killed just after 
development of the erosion to confirm a damage of the skin. Whole epithelium was 
damaged, but a bottom of the hair follicles was intact, suggesting that phenol does not 
penetrate so much deep in the skin. Then we performed chemical peeling on some female 
participants using 89% phenol as a reagent and found that it was well-tolerated. (The tird 
paragraph , Page 5).
Q5 In M and M. All patients with ASC-US+ and HPV+ had CIN1+. Is it right? If so, it 
sounds unusual. No cases without CIN lesions under colposcopy in this study?
A5. This is reasonable question. To answer to this question, we revised Fig.3 and the 
manuscript. The entry numbers increased as 509 subjects, since we realized that five cases 
were omitted, since they showed negative results after histological evaluation under 
colposcopy. This was revised in the M&M. 
Q6 Line 171-172. The changes in diagnosis are down/upgraded, but it is not described 
clearly enough. Many cases would be downgraded during the treatment anyway, and the 
initial diagnosis won't be affected. What does this mean?
A6; This comment made readers confused, and therefore it was omitted. 
Q7 Is it the best way to show the number of treatments in the median and 95% CI? Line 
188-
A6. We modified this is average and 95% CI.

Reviewer 2 Report

Comments and Suggestions for Authors

Dear authors, thank you for your precious work. There are some issues to be clarified before considering for publication.

line 59 I would not support including AIS, since there is a higher chance of invasive cancer at diagnosis compared to hsil.

line 83 to 88: not pertinent in intro, move to results or conclusion

Line 96: change to LSIL or HSIL, according to LAST 2012 terminology

Line 118 to 124 move to results, not pertinent in M&M

Please a little clarification about Materials and methods: the lesions included in the study were located on the exocervix? the ones involving the endocervix (thus impossibile to treat) were excluded? please provide more details on this important confounder

Please improve readability of the figures provided

I would expand discussion on treatment of VAIN, please you can refer to recent multi society statement: Kesic, Vesna MD1; Carcopino, Xavier MD2; Preti, Mario MD3; Vieira-Baptista, Pedro MD4,5; Bevilacqua, Federica MD3; Bornstein, Jacob MD6; Chargari, Cyrus MD7; Cruickshank, Maggie MD8; Erzeneoglu, Emre MD9; Gallio, Niccolò MD3; Gultekin, Murat MD10; Heller, Debra MD11; Joura, Elmar MD12; Kyrgiou, Maria MD13,14; Madić, Tatjana MD15; Planchamp, François MD16; Regauer, Sigrid MD17; Reich, Olaf MD18; Esat Temiz, Bilal MD19; Woelber, Linn MD19,20; Zodzika, Jana MD21; Stockdale, Colleen MD22. The European Society of Gynaecological Oncology (ESGO), the International Society for the Study of Vulvovaginal Disease (ISSVD), the European College for the Study of Vulval Disease (ECSVD), and the European Federation for Colposcopy (EFC) Consensus Statement on the Management of Vaginal Intraepithelial Neoplasia. Journal of Lower Genital Tract Disease 27(2):p 131-145, April 2023. | DOI: 10.1097/LGT.0000000000000732

Comments on the Quality of English Language

Minor editing

Author Response

#Reviewer-2
Dear authors, thank you for your precious work. There are some issues to be clarified 
before considering for publication.
Q1. line 59 I would not support including AIS, since there is a higher chance of invasive 
cancer at diagnosis compared to hsil.
A1. This is important question. AIS was found in 3 cases of CIN3 during treatment, and 
all these lesions disappeared by this therapy. We added some comments as follows; 
Adenocarcinoma in situ was also present in three patients with CIN3; these patients had 
AIS coexisted with CIN3 and resolved during therapy. (The first paragraph, Page 7)
Q2. line 83 to 88: not pertinent in intro, move to results or conclusion
A2. These comments are omitted according to this suggestion.
Q3. Line 96: change to LSIL or HSIL, according to LAST 2012 terminology
A3. This part is a mistake, and this is revised. 
Q4. Line 118 to 124 move to results, not pertinent in M&M
A4. We transferred these comments into Result section. 
Q5. Please a little clarification about Materials and methods: the lesions included in the 
study were located on the exocervix? the ones involving the endocervix (thus 
impossibile to treat) were excluded? please provide more details on this important 
confounder.
A5. To answer to this question we added some comments as follows; In cases where the 
lesions extended into the cervical canal, the tip was dipped in liquid phenol and inserted 
into the canal. However, some women whose lesions were not well visible had undergone 
surgical treatment (The third paragraph, Page 7).
Q6. Please improve readability of the figures provided
A6. We modified it as much as we could.
Q7. I would expand discussion on treatment of VAIN, please you can refer to recent 
multi society statement: Kesic, Vesna MD1; Carcopino, Xavier MD2; Preti, Mario 
MD3; Vieira-Baptista, Pedro MD4,5; Bevilacqua, Federica MD3; Bornstein, Jacob 
MD6; Chargari, Cyrus MD7; Cruickshank, Maggie MD8; Erzeneoglu, Emre MD9; 
Gallio, Niccolò MD3; Gultekin, Murat MD10; Heller, Debra MD11; Joura, Elmar 
MD12; Kyrgiou, Maria MD13,14; Madić, Tatjana MD15; Planchamp, François MD16; 
Regauer, Sigrid MD17; Reich, Olaf MD18; Esat Temiz, Bilal MD19; Woelber, Linn 
MD19,20; Zodzika, Jana MD21; Stockdale, Colleen MD22. The European Society of 
Gynaecological Oncology (ESGO), the International Society for the Study of 
Vulvovaginal Disease (ISSVD), the European College for the Study of Vulval Disease 
(ECSVD), and the European Federation for Colposcopy (EFC) Consensus Statement on 
the Management of Vaginal Intraepithelial Neoplasia. Journal of Lower Genital Tract 
Disease 27(2):p 131-145, April 2023. | DOI: 10.1097/LGT.0000000000000732
A7. This is a precious information. We cited this as reference #21, and added some 
comments in the discussion as follows; 
Guidelines by several groups, including the International Society for the Study of 
Vulvovaginal Disease, do not recommend TCA therapy (a kind of chemical peeling 
thereapy) for vulvo-vaginal diseases (21). In the present study, the average numbers of 
treatment sessions required for disease clearance were 4.8 and 6.7 for VaIN1 and VaIN2, 
respectively, which did not differ from those needed for CIN1 and CIN2 (p > 0.1, MannWhitney U test). The clearance rates were 100% (13/13) and 83% (5/6) for cases with 
VaIN1 and VaIN2, respectively, suggesting that this treatment method is effective for 
VaINs1/2.(). (The second paragraph, Page 14)

Round 2

Reviewer 2 Report

Comments and Suggestions for Authors

I support accepting the present paper